# Serotonin Pathway of Tryptophan Metabolism in Small Intestinal Bacterial Overgrowth—A Pilot Study with Patients Diagnosed with Lactulose Hydrogen Breath Test and Treated with Rifaximin

**DOI:** 10.3390/jcm10102065

**Published:** 2021-05-12

**Authors:** Cezary Chojnacki, Tomasz Popławski, Paulina Konrad, Michal Fila, Jan Chojnacki, Janusz Błasiak

**Affiliations:** 1Department of Clinical Nutrition and Gastroenterological Diagnostics, Medical University of Lodz, 90-647 Lodz, Poland; cezary.chojnacki@umed.lodz.pl (C.C.); paulina.konrad@umed.lodz.pl (P.K.); 2Department of Molecular Genetics, Faculty of Biology and Environmental Protection, University of Lodz, 90-236 Lodz, Poland; tomasz.poplawski@biol.uni.lodz.pl; 3Department of Developmental Neurology and Epileptology, Polish Mother’s Memorial Hospital Research Institute, 93-338 Lodz, Poland; michalfila@poczta.onet.pl

**Keywords:** small intestinal bacterial overgrowth, SIBO, tryptophan metabolism, serotonin, rifaximin

## Abstract

Small intestinal bacterial overgrowth (SIBO) is a condition associated with diverse clinical conditions and there is no gold standard in its diagnosis and treatment. Tryptophan (Trp) metabolism may be involved in etiology of gastrointestinal diseases and is regulated by intestinal microbiota. In our study we investigated aspects of the serotonin (5-HT) pathway of Trp metabolism in three groups of individuals based on the hydrogen concentration in the lactulose hydrogen breath test (LHBT): controls (<20 ppm) and SIBO patients (≥20 ppm), with diarrhea (SIBO-D) or constipation (SIBO-C). The SIBO-D patients showed an increased serum concentration of 5-HT and small intestinal mucosa mRNA expression of tryptophan hydroxylase 1 (TPH-1), a rate-limiting enzyme in 5-HT biosynthesis. Urinary 5-hydroxyindoleacetic acid (5-HIAA), the main metabolite of 5-HT, was higher in both group of SIBO patients than controls. A positive correlation between 5-HIAA and LHBT was observed. A two-week treatment with rifaximin decreased hydrogen in LHBT and 5-HIAA concentration in SIBO patients. In conclusion, the serotonin pathway of Trp metabolism may play a role in the pathogenesis of hydrogen-positive SIBO and it may influence the diversification of SIBO into variants with diarrhea or constipation. As urinary 5-HIAA concentration correlates with LHBT, TPH-1 expression in colonic mucosa and TH-5 in serum of SIBO patients, it can be considered as a non-invasive marker of this condition.

## 1. Introduction

Small intestinal bacterial overgrowth (SIBO) is a disorder with no distinct clinical manifestations as it is associated with abdominal pain/discomfort, bloating, flatulence, diarrhea, and constipation that all may be linked to several other syndromes in the gastrointestinal (GI) tract, including irritable bowel syndrome (IBD) (reviewed in [1]). Therefore, a precise definition of this disease in the clinical context may lack specificity and consistency. At the cellular level, this disorder is currently defined as occurring with increase in bacterial flora equal to or greater than 10^3^ colony-forming units (CFU) per mL, but previously it was postulated even 10^5^ CFU/mL of upper gut aspirate [1,2,3]. However, we still do not know what bacterial population in the GI tract should be considered as a “normal”. Literature data suggest that such normal population should rather not exceed 10^2^ CFU/mL and 10^5^ CFU/mL should be attributed to anatomical abnormalities in the GI tract [4]. Original clinical definition of SIBO was associated with the manifestation of maldigestion and malabsorption related to qualitative and/or quantitative changes in the small intestinal microbiota (reviewed in [5]). Today, SIBO is rather perceived as the condition underlying a broad range of intestinal and extraintestinal diseases, but it many cases it is unclear whether SIBO is a cause, a consequence, or just an epiphenomenon in relation to the other GI tract disorder [6]. Several risk factors, including older age, female sex, steatorrhea, and the use of opioids as well as several other coexisting diseases, can be considered in SIBO pathogenesis, but due to association of SIBO with other disorders, it is not easy to distinguish risk factor specific for that disease (reviewed in [7]). Currently applied diagnostic tests suffer from serious limitations, as they largely either invasive or unreliable [6,8]. Some authors link SIBO with inflammation and consequently, inflammatory markers, such as fecal calprotectin, can be used to diagnose the disease (reviewed in [9]). Some nutritional disturbances, including increased folate and vitamin B12 deficiency, resulting from bacterial synthesis of folic acid, can be helpful in SIBO diagnosis ([6]).

Determination of the population of bacterial flora in a sample of fluid aspirated from small bowel intestine culture is believed to be the gold standard in SIBO diagnosis (reviewed in [10]). As this method is invasive, time-consuming, and relatively expensive, other diagnostic approaches have been applied, first of all breath tests, but they suffer from lower sensitivity and specificity as compared with aspiration and culture. SIBO therapy is mainly based on the use of antibiotics, but their side effects, development of multi-resistant strains and the potential of eradication of beneficial bacteria, imply the need for searching for new therapeutic modalities (reviewed in [10]). Of the antibiotic applied in SIBO treatment, rifaximin is preferentially used, especially in patients with methane-positive SIBO breath tests (reviewed in [11]). Many other antibiotic regimens are prescribed for SIBO patients, but similarly to the definition and diagnosis, there is no generally accepted consensus in SIBO treatment. Several other aspects of SIBO need explanation, including determination of the reason that some SIBO patients present the disease with diarrhea and others—with constipation. Therefore, studies on molecular mechanisms of SIBO pathogenesis are justified.

Tryptophan (Trp), an essential exogenous amino acid in humans, plays an important role in the GI tract homeostasis and disturbances in its metabolism are associated with intestinal diseases [12]. The vast majority of Trp is absorbed in the small intestine and the rest of it may be a metabolic substrate for colonic microbes so potential association of Trp metabolism with SIBO pathogenesis is rational (reviewed in [13]). Tryptophan metabolism in hosts proceeds via the kynurenine pathway or serotonin pathway to yield active metabolites. In gut microbes Trp is metabolized in the indole pathway (reviewed in [13]). Microbial Trp catabolites resulting from proteolysis influence host health (reviewed in [14]). Serotonin (5-hydroxytryptamine, 5-HT) is in 90% produced from Trp in the distal GI tract by enterochromaffin cells and serotogenic neurons of the myenteric plexus expressing tryptophan hydroxylase 1 (TPH-1) and 2, respectively, which are rate limiting enzymes in 5-HT biosynthesis Boadle 93. Remaining 10% is produced by serotonergic neurons in the central nervous system by TPH2. Once released, 5-HT is inactivated by the serotonin reuptake transporter and broken down into 5-hydroxyindole acetic acid (5-HIAA), finally excreted with urine [15].

Recently we have shown that patients with lymphocytic colitis (LC), a colon disease sharing some clinical manifestations with SIBO, exhibited altered 5-HT metabolism [16]. In the present work we analyzed some components of 5-HT metabolism in SIBO patients with diarrhea or constipation as compared with healthy individuals. The analysis included TPH1 mRNA expression in the small intestinal mucosa as well as the levels of 5-HT in serum and 5-HIAA in urine of all individuals enrolled in this study. All SIBO patients were treated for two weeks treatment with rifaximin and LHBT and 5-HIAA determination were performed again.

## 2. Materials and Methods

### 2.1. Patients and Clinical Examination

The study included 120 individuals recruited from the Department of Gastroenterology, Medical University of Lodz, Lodz, Poland, in 2009–2017. They were divided into three groups, 40 individuals each: healthy subjects (group I, controls), patients with small intestinal bacterial overgrowth and chronic diarrhea (group II, SIBO-D), and 40 patients with SIBO and constipation (group III, SIBO-C). The SIBO-D group was characterized by loose or watery stools, occurring >25% of the time, at least for six months. In the SIBO-C group there were two or fewer bowel movements a week and hard and lumpy stools for minimum six months. The hydrogen breath test was performed to diagnose SIBO using Gastrolyzer (Bedfont, Ltd., Harrietsham, UK). An increase in hydrogen to more than or equal to 20 ppm by 90 min during the test was the criterion for SIBO occurrence [17]. To determine other organic diseases of the GI tract, all subjects underwent endoscopic and histological examination of duodenal, small intestinal, and colonic mucosa. The exclusion criteria were ulcerative and lymphocytic colitis, Crohn disease, allergy and food intolerance, liver and renal disease, diabetes, carcinoid tumor, and the use of antibiotics, probiotics, and psychotropic drugs in the month prior to enrollment.

The study was conducted in accordance with the Declaration of Helsinki and the principles of Good Clinical Practice. Written consent was obtained from each subject enrolled in the study and the study protocol was approved by the Bioethics Committee of Medical University of Lodz (permit number RNN/11/17/KB dated 16 February 2017).

### 2.2. Laboratory Tests

Blood cell count, profile of proteins and lipids, glucose, bilirubin, iron, urea, creatinine, free thyroxine, free triiodothyronine, alanine and asparagine aminotransferase, gamma-glutamyltranspeptidase, alkaline phosphatase, amylase, lipase, and fecal calprotectin were determined in all subjects enrolled in this study.

Venous blood and 24 h urine collection samples were centrifuged and stored at −70 °C. The serum concentration of C-reactive protein (CRP) was determined by a latex agglutination photometric assay in COBAS INTEGRA 800 (Roche Diagnostic, Basel, Switzerland), the fecal calprotectin (FC) was evaluated by a sandwich ELISA test in Quantum Blue Reader (Buhlmann Diagnostics, Amherst, NH, USA), and the serum 5-HT and urine 5-HIAA levels were determined with ELISA tests, number 59,121 and 59,131, respectively. The obtained results of 5-HIAA were converted from nanogram/mL to microgram/24 h. All participants were recommended to have the same balanced diet, with a total energy value of 1800 kcal and with 1100 mg of tryptophan, three days before and two weeks during drug treatment. The energy value and the tryptophan intake were calculated using the Kcalmar.pro-Premium application (Hermex, Lublin, Poland).

The level of mRNA expression was determined by RT-PCR with 50 mg of mucosa of distal part of small intestine for each individual enrolled in this study. Briefly, colonic tissues were rapidly permeated to stabilize and protect cellular RNA with RNA stabilization reagent RNAlater^®^ (Qiagen, Hilden, Germany). Prior to isolation of total RNA, fragments of colonic tissues were homogenized with TissueRuptor (Qiagen). Then, total RNA was isolated using Qiagen RNeasy Plus Mini Kit (Qiagen). The quantity and quality of RNA were estimated spectrophotometrically by Take3 plate on Synergy HT Microplate Reader (BioTek Instruments, Winooski, VT, USA). The real-time gene expression analysis was performed using the TaqMan Gene Expression Assays (Thermo Fisher Scientific, Waltham, MA, USA) with probes specific for TPH-1 and SensiFASTTM Probe No-ROX One-Step Kit (Bioline, Taunton, MA, USA). The hypoxanthine phosphoribosyltransferase (HPRT, Assay ID: Hs01003267_m1) gene was a reference. Real-time PCR was performed with BioRad CFX96 thermal cycler (BioRad, Hercules, CA, USA). Expression analysis was performed with CFX Manager 1.6 software (BioRad, Hercules, CA, USA) with the ΔΔCt method [18].

### 2.3. Rifaximin Treatment

All SIBO patients were recommended to take rifaximin for 14 days at total daily dose 1200 mg divided into three doses 400 mg each. They were also recommended to keep on the diet as above. Four weeks after rifaximin administration, the LHBT test was performed and the urinary 5-HIAA excretion was determined again.

### 2.4. Data Analysis

Shapiro–Wilk W-test was used to check whether the data followed the normal distribution. The homogeneity of variances within three groups were checked by Levene’s test. As they were not homogenous, the variability within these group were analyzed by Kruskal–Wallis test and the difference between means were compared with Scheffe’s multiple comparison test. The differences between two groups were determined using Student’s *t*-test or Mann–Whitney U-test. The Wilcoxon matched pairs signed rank test was used to assess the difference between each set of matched pairs before and after rifaximin treatment. Correlations between the quantitative variables were analyzed using Spearman’s correlation coefficient. All regression lines were drawn by the least square methods. All statistical analyses were performed with the Statistica 13.3 software (TIBCO Software Inc., Palo Alto, CA, USA).

## 3. Results

All individuals enrolled in this study were divided into two groups based on the exhaled hydrogen concentration in the LHBT test: controls (<20 ppm) and SIBO patients (≥20 ppm). The latter were further divided into two subgroups in dependence on the associated syndromes diarrhea (SIBO-D) or constipation (SIBO-C) (Figure 1).

A pronounced (*p* < 0.001) increase in breath hydrogen was observed in SIBO patients in LHBT.

There was no significant difference in the distribution of gender, age, BMI, ALT, and AST activity as well as glomerular filtrating ratio between SIBO patients and controls (Table 1). The concentration of the C-reactive protein was higher (*p* < 0.001) in SIBO-D patients than in controls. The concentration of fecal calprotectin was higher in both SIBO-D (*p* < 0.001) and SIBO-C (*p* < 0.05) patients.

The SIBO-D patients displayed an increased small intestine mucosa expression of tryptophan hydroxylase 1, a rate-limiting enzyme in 5-HT synthesis, mRNA as compared with controls (*p* < 0.001) and the other group of SIBO patients, SIBO-C (*p* < 0.001) (Figure 2, upper graph). The same relationship was for serum concentration of 5-HT (Figure 2, middle graph). Urinary 5-HIAA concentration was higher in both group of SIBO patients than controls (*p* < 0.001) and SIBO-D patients displayed higher 5-HIAA concentration than their SIBO-C counterparts (*p* < 0.001) (Figure 2, lower graph).

A positive correlation was found between urinary 5-HIAA concentration and hydrogen in LHBT test in both SIBO-D and SIBO-C patients (*p* < 0.05 for either group, Figure 3).

A positive correlation was observed between the urinary 5-HIAA excretion and TPH-1 mRNA expression in small intestine mucosa of SIBO-C patients (*p* < 0.01, Figure 4).

Rifaximin was well tolerated and four weeks after its administration, LHBT was performed and 5-HIAA in urine was determined in all SIBO patients.

Rifaximin treatment resulted in a decrease in the average concentration of hydrogen in LHBT in both SIBO groups (*p* < 0.001, Figure 5, upper graph), but average LHBT values were above 20 ppm and only 12 patients in either group presented values below 20 ppm. Similarly, urinary 5-HIAA decreased after the treatment in both groups (*p* < 0.001, Figure 5, lower graph).

We observed a weak/moderate positive correlation between changes in urine 5-HIAA concentration and changes in serum concentration of 5-HT in SIBO-D patients after rifaximin treatment (*p* < 0.01, Figure 6).

## 4. Discussion

Small intestine bacterial overgrowth presents a problem in the gastrointestinal clinic, as there are not a consensus concerning the definition of SIBO and so its diagnostic criteria as well as therapeutic procedures. Tryptophan metabolism is involved in the pathogenesis of several gastrointestinal disorders, but little is known on Trp role in SIBO etiology [12,13,14]. Moreover, nutritional implications of SIBO are not completely known [8]. That is why we took a closer look at the serotonin pathway of Trp metabolism in SIBO patients in association with SIBO diagnosis and therapy. We enrolled subjects with recommended diet supplemented with Trp and investigated some products of its metabolism in small intestine colonic mucosa, serum and urine.

In this study, we employed lactulose breath test, which recently was reported to have lower susceptibility, specificity and worse general performance as compared with its glucose counterpart on the basis of a meta-analysis [19]. However, lactulose test seems to be more reliable when intestinal motility is impaired [17,20]. Furthermore, glucose in contrary to lactulose, tends to be absorbed by the duodenum and jejunum, impeding microbiota assessment in distal parts of the small intestine. Therefore, we limited our consideration to overgrowth caused by hydrogen-producing microorganism. However, many pathogenic microorganism can contribute to intestinal overgrowth in SIBO patients, including *E. coli*, *Enterococcus,* and *Klebsiella*, but also *Methanobrevibacter smithii* may significantly influence intestinal microbiota and be detected by methane rather and not hydrogen breath test [21]. The use a hydrogen breast test not supplemented with its methane counterpart is a limitation of our study but, on the other hand, this limitation increased the specificity of our work to hydrogen-positive SIBO. Therefore, study may be extended by research on methane-positive SIBO.

Constipation is not a typical sign of SIBO and positive results in hydrogen test can be underlined by a delayed intestinal transit in constipation patients. However, the only reliable method to check the presence/absence of SIBO would be invasive analysis of the number of bacteria or/and composition of bacteria in aspiration fluid from the small intestine. This is a limitation of our study, but further studies are required to confirm whether constipation can be a SIBO-associated symptom.

We enrolled a moderate number of patients in our study. However, we applied a strong selection of individuals to minimize the influence of confounding factors, such as sex, age, BMI, and comorbidities, including irritable bowel syndrome, which in fact made us to work with much larger group of subjects.

Alterations of Trp metabolism pathways have been also described in IBS patients (reviewed in [22]). These alterations may be underlined by changes in the intestinal microflora, which metabolizes excessive dietary Trp and produces many Trp metabolites that are more active than the parental compound (reviewed in [23]). They may be linked with excessive immune activation of the host and progression/outcome of various diseases, including IBS (reviewed in [24]). SIBO is highly prevalent in IBS—Its prevalence calculated in a recent meta-analysis was about 31% [21,25]. This meta-analysis included studies with different SIBO definitions and different methods to its diagnosis as well as different control groups—Healthy subject and patients with non-IBD diseases. Our study shows an association between changes in Trp metabolism and SIBO occurrence—If such changes are associated with IBS, they could be still underlined by SIBO, unless the latter is excluded. To determine what is a primary reason of changes in Trp metabolism that are associated with several clinical symptoms requires additional cause-effects molecular studies.

Initial biochemical characterization of SIBO patients showed an increase in CRP and FC as compared with controls (Table 1). Fecal calprotectin is frequently considered as a marker of SIBO, especially when this disorder is associated with some mental disturbances, reflecting a potential involvement of the gut–brain axis in SIBO pathogenesis [9]. In turn, increased CRP as a marker of inflammation, confirms inflammatory aspects of SIBO as it can evoke an inflammatory reaction in the intestinal mucosa, potentiating main symptoms of the disease, although no correlation was found between SIBO severity and inflammation in obese children [26,27,28].

We observed an increased in colonic mucosa TPH-1 mRNA expression and in serum 5-HT concentration in SIBO-D patients as compared with controls and their SIBO-C counterparts (Figure 2). Therefore, TPH-1 and 5-HT may be considered as potential markers differentiating SIBO-D from SIBO-C, and factors, which may contribute to mechanism leading to different SIBO outcomes in the categories of bowel movement and stool consistency. The concentration of 5-HIAA in urine was higher in SIBO-D and C patients as compared with controls (Figure 2). On the other hand, SIBO-D patients showed a higher urine 5-HIAA concentration than their SIBO-C counterparts. In summary, we observed an increased level of serotonin and its 5-HIAA metabolite in SIBO patients that might result from an increased activity of TPH-1, a rate-limiting enzyme in serotonin synthesis. Moreover, the concentration of 5-HT and 5-HIAA as well as TPH-1 expression may distinguish SIBO-D from SIBO-C patients.

We observed a positive correlation between 5-HIAA and results of breath test used for SIBO diagnosis in both groups of patients (Figure 3). 5-hydroxyindoleacetic acid, in contrary to TPH-1 and 5-HT, was determined in a non-invasive procedure. Increased values of TPH-1, 5-HT, and 5-HIAA inspired us to look for a relationship between 5-HIAA and remaining two elements of Trp metabolism. We found positive correlations between 5-HIAA and TPH-1 and 5-HT (Figure 4 and Figure 5). These results suggest that 5-HIAA can be used in parallel or a substitute of LHBT, TPH-1, and 5-HT in analytical analysis in SIBO patients.

Although there is no generally accepted consensus on SIBO therapy, rifaximin is frequently used [29]. We observed that two weeks administration of 400 mg rifaximin three times daily resulted in a significant decrease in LHBT values, but the average concentration of expired hydrogen was still above the SIBO cut-off level (20 ppm) (Figure 6). Similarly, the average urine 5-HIAA concentration decreased after rifaximin treatment in both SIBO groups, but it did not reach the respective average values for the control group (Figure 6). We determined neither TPH-1, nor 5-HT after rifaximin treatment, as there was no justification for repeating so invasive procedures in SIBO patients. We observed a positive correlation between rifaximin treatment-related changes in 5-HIAA concentration and hydrogen concentration in LHBT (Figure 6).

In conclusion, the serotonin pathway of tryptophan metabolism may be changed in patients with SIBO diagnosed on the basis of lactulose breath test, but this should be confirmed by randomized clinical trials (RCTs). As we showed that urinary 5-HIAA positively correlated with other Trp metabolites, it could be considered as a non-invasive marker in hydrogen-positive SIBO but, again, randomized clinical trials are needed to verify this speculation. Rifaximin may lower the intensity of hydrogen-specific SIBO by lowering microorganism cultures in the GI tract resulting in lower values of LHBT and 5-HIAA concentration. The outcome of rifaximin treatment may not depend on whether SIBO is associated with diarrhea or constipation. Further research on the role of tryptophan metabolism in SIBO pathogenesis should include methane-specific variant of the disease, other metabolites of the serotonin pathways as well as products of the other tryptophan metabolism pathway—The kynurenine pathway. The occurrence of SIBO with associated diarrhea or constipation should be also addressed in these studies.

Finally, we presented the results of studies on SIBO patients diagnosed with lactulose hydrogen breath test and treated with rifaximin, but our pilot study should be confirmed in RCTs.

## Figures and Tables

**Figure 1 jcm-10-02065-f001:**
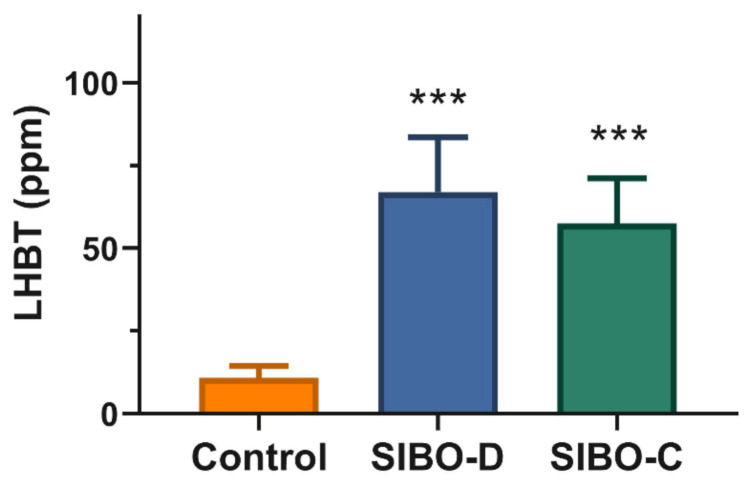
Hydrogen in 90 min lactulose hydrogen breath test (LHBT) in small intestine bacterial overgrowth (SIBO) patients with diarrhea (SIBO-D) or constipation (SIBO-C) and control individuals. *n* = 40 in each group, error bars indicate SD, *** *p* < 0.001 as compared with controls.

**Figure 2 jcm-10-02065-f002:**
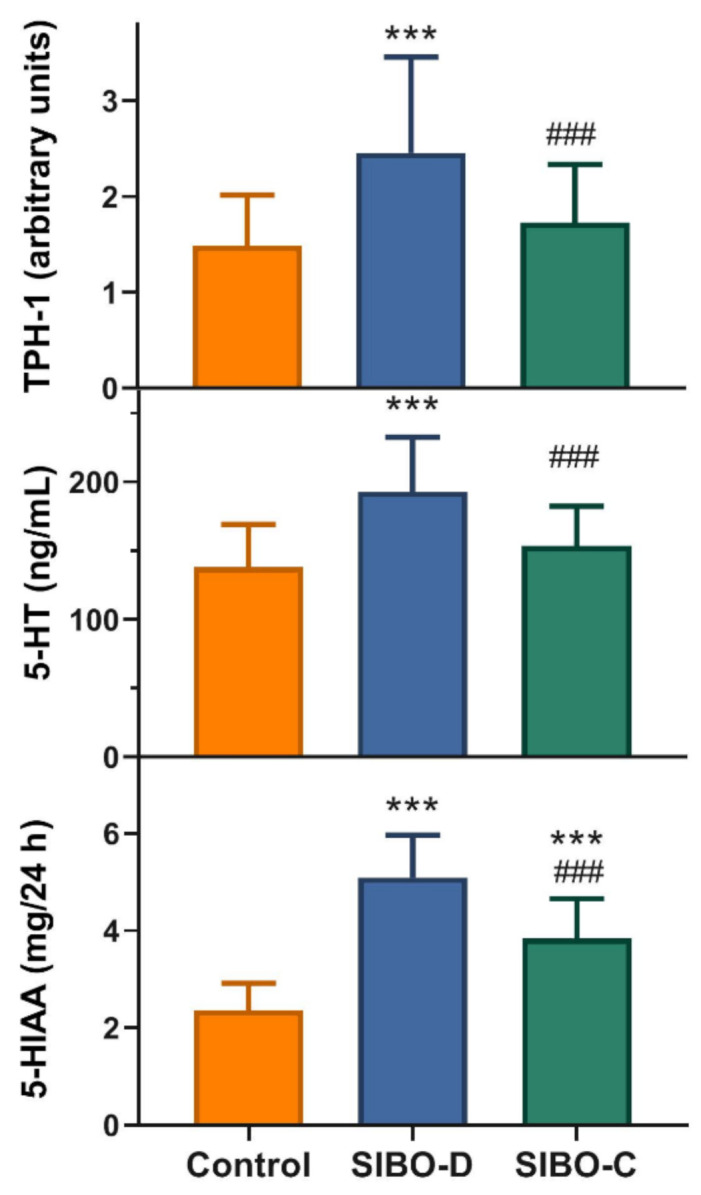
Serotonin metabolism in small intestine bacterial overgrowth (SIBO) patients with diarrhea (SIBO-D) or constipation (SIBO-C) and control individuals. Tryptophan hydroxylase 1 (TPH-1) mRNA expression in small intestine mucosa determined with the hypoxanthine phosphoribosyltransferase gene as a reference (A). Serum concentration of serotonin (5-HT) (B). Urinary concentration of 5-hydroxyindoleacetic acid (5-HIAA) after a 24-h collection. Mean + SD, *n* = 40 in each group; ***, *p* < 0.001 as compared with controls; ###, *p* < 0.001 as compared with SIBO-D patients.

**Figure 3 jcm-10-02065-f003:**
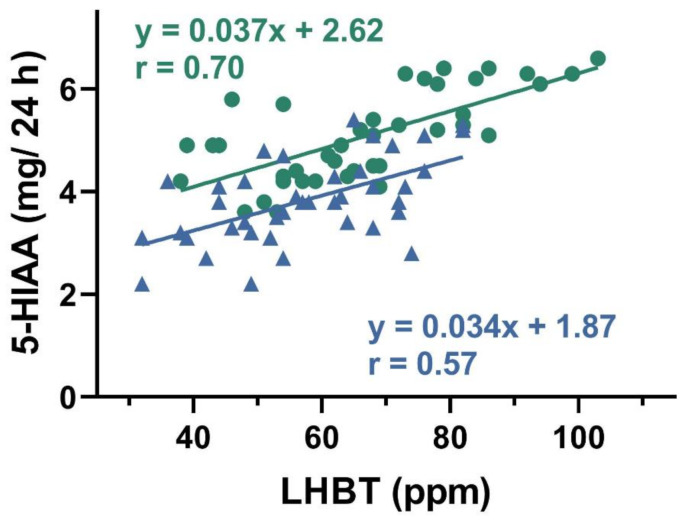
Correlation between exhaled hydrogen (ppm) in lactulose hydrogen breath test (LHBT) in small intestine bacterial overgrowth (SIBO) patients with diarrhea (green, *n* = 40) or constipation (navy blue, *n* = 40)) and urinary concentration of 5-hydroxyindoleacetic acid (5-HIAA) collected in 24 h. The Spearman rank correlation coefficient (r) was used to evaluate of the strength of correlation. The regression lines were drawn using the least squares method.

**Figure 4 jcm-10-02065-f004:**
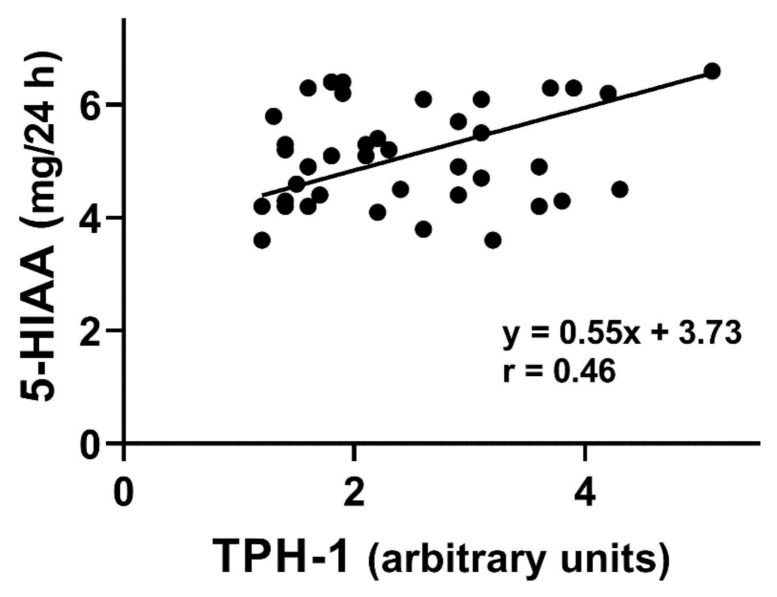
Correlation between tryptophan hydroxylase 1 (TPH-1) mRNA expression in small intestine mucosa in small intestine bacterial overgrowth (SIBO) patients with constipation (*n* = 40) and urinary concentration of 5-hydroxyindoleacetic acid (5-HIAA). The Spearman rank correlation coefficient (r) was used to evaluate of the strength of correlation. The regression line was drawn using the least squares method.

**Figure 5 jcm-10-02065-f005:**
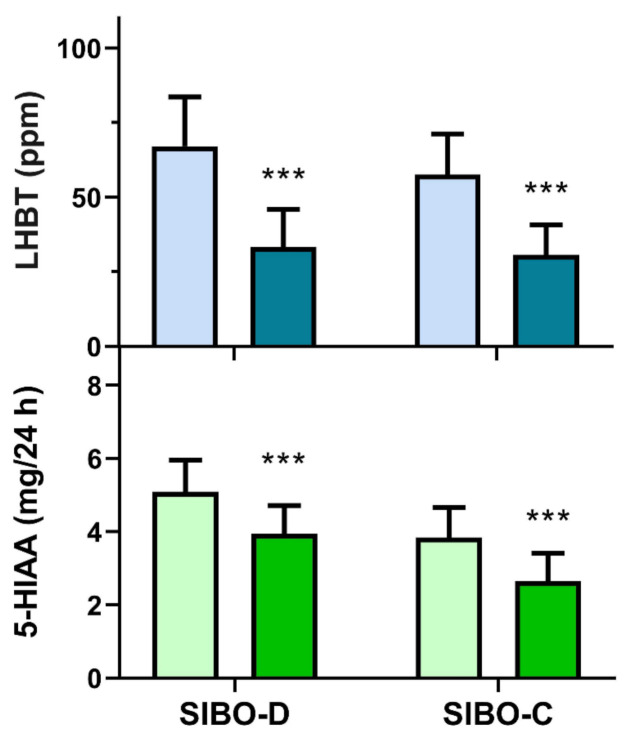
Hydrogen (ppm) in 90 min lactulose hydrogen breath test (LHBT) in small intestine bacterial overgrowth (SIBO) patients with diarrhea (SIBO-D) or constipation (SIBO-C) and urinary concentration of 5-hydroxyindoleacetic acid (5-HIAA) after a 24 h collection before (light colors) and after (dark colors) 14 days treatment with rifaximin at 1200 mg daily for 14 consecutive days. Mean + SD, *n* = 40 in either group, *** *p* < 0.001.

**Figure 6 jcm-10-02065-f006:**
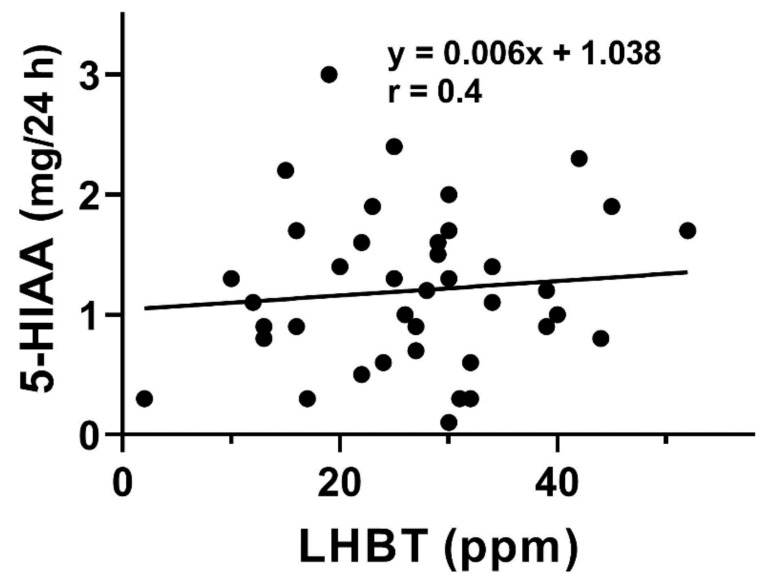
Correlation between serum concentration of serotonin (5-HT) in small intestine bacterial overgrowth (SIBO) patients with diarrhea (*n* = 40) and urinary concentration of 5-hydroxyindoleacetic acid (5-HIAA). The Spearman rank correlation coefficient (r) was used to evaluate of the strength of correlation. The regression line was drawn using the least squares method.

**Table 1 jcm-10-02065-t001:** Characteristics of small intestine bacterial overgrowth (SIBO) patients with diarrhea (SIBO-D) or constipation (SIBO-C) and control individuals enrolled in this study.

Feature ^a^	Controls	SIBO-D Patients	SIBO-C Patients
Number	40	40	40
Gender (M/F)	18/22	16/24	14/26
BMI (kg/m^2^)	23.1 ± 1.6	22.6 ± 0.8	24.1 ± 2.3
Age (years)	43.4 ± 5.1	41.2 ± 7.4	45.3 ± 10.4
ALT (aU/L)	14.3 ± 3.8	18.4 ± 8.3	19.4 ± 12.3
AST (aU/L)	12.2 ± 3.1	16.1 ± 8.6	17.3 ± 7.2
CRP (mg/L)	1.24 ± 0.32	6.14 ± 1.28 ***	3.92 ± 1.65
FC (µg/g)	21.5 ± 7.41	46.8 ± 11.3 ***	34.6 ± 10.4 *
GFR (mL/min)	104.0 ± 10.3	96.5 ± 14.3	88.4 ± 10.5

^a^ average ± SD; SD, standard deviation; M, male; F, female; BMI, body mass index; ALT, alanine aminotransferase; ASP, asparagine aminotransferase; CRP, C-reactive protein; FC, fecal calprotectin; GFR, GFR—glomerular filtrating ratio; aU, arbitrary unit; *, *p* < 0.05; and ***, *p* < 0.001 as compared with healthy controls.

## Data Availability

The data that support the findings of this study are available on request from the corresponding author. The data are not publicly available due to privacy and ethical restrictions and are stored at www.umed.lodz.pl (accessed on 1 May 2021).

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
