# Peer review of "Serotonin Pathway of Tryptophan Metabolism in Small Intestinal Bacterial Overgrowth—A Pilot Study with Patients Diagnosed with Lactulose Hydrogen Breath Test and Treated with Rifaximin"

_jcm, 2021, doi:10.3390/jcm10102065_

Round 1

Reviewer 1 Report

In this original article Chojnacki et al demonstrated that, in patients with SIBO, serum levels of 5HT and urinary 5HIAA are higher than in controls. Additionally, antibiotic therapy for SIBO is able to reverse such tryptophan metabolism imbalance. Main comments:

1) Page 1: exponents for CFU should be in superscript.

2) Why did Authors use lactulose instead of glucose breath test, considering that it has an inferior diagnostic performance? (see Losurdo G et al, J Neurogastroenterol Motil 2020). Please discuss.

3) Constipation is not a typical sign of SIBO. Could these patients have a delayed intestinal transit that has been misinterpreted as SIBO by LHBT?

4) Were patients with carcinoid tumor ruled out?

5) Page 8, regarding the correlation in figure 6, Authors are not allowed to draw such conclusion because an r of 0.4 is very low.

6) Alteration of tryptophan pathways has been described in IBS patients as well.  Considering that in several cases IBS is related with SIBO, this observation should be analyzed and discussed.

Author Response

In this original article Chojnacki et al demonstrated that, in patients with SIBO, serum levels of 5HT and urinary 5HIAA are higher than in controls. Additionally, antibiotic therapy for SIBO is able to reverse such tryptophan metabolism imbalance. Main comments:

Comment: 1) Page 1: exponents for CFU should be in superscript.

Answer: We have corrected that.

Comment 2) Why did Authors use lactulose instead of glucose breath test, considering that it has an inferior diagnostic performance? (see Losurdo G et al, J Neurogastroenterol Motil 2020). Please discuss.

Answer: We think that there is no consensus in this matter. We have added the following fragment to Discussion:

“In this study, we employed lactulose breath test, which recently was reported to have lower susceptibility, specificity and worse general performance as compared with its glucose counterpart on the basis of a meta-analysis [19]. However, lactulose test seems to be more reliable when intestinal motility is impaired [17,20]. Furthermore, glucose in contrary to lactulose, tends to be absorbed by the duodenum and jejunum, impeding microbiota assessment in distal parts of the small intestine.”

with new references:

  1. Losurdo, G.; Leandro, G.; Ierardi, E.; Perri, F.; Barone, M.; Principi, M.; Leo, A.D. Breath Tests for the Non-invasive Diagnosis of Small Intestinal Bacterial Overgrowth: A Systematic Review With Meta-analysis. Journal of neurogastroenterology and motility 2020, 26, 16-28, doi:10.5056/jnm19113.
  2. Pimentel, M. Breath Testing for Small Intestinal Bacterial Overgrowth: Should We Bother? The American journal of gastroenterology 2016, 111, 307-308, doi:10.1038/ajg.2016.30.

Comment: 3) Constipation is not a typical sign of SIBO. Could these patients have a delayed intestinal transit that has been misinterpreted as SIBO by LHBT?

Answer: We have added the following fragment to Discussion:

“Constipation is not a typical sign of SIBO and positive results in hydrogen test can be underlined by a delayed intestinal transit in constipation patients. However, the only reliable method to check the presence/absence of SIBO would be invasive analysis of the number of bacteria or/and composition of bacteria in aspiration fluid from the small intestine. This is a limitation of our study.”

Comment: 4) Were patients with carcinoid tumor ruled out?

Answer: Yes – no one had facial flushing, asthma-like symptoms, skin changes, sweating, unexplained weight gain and diarrhea was chronic, not paroxysmal. We have added carcinoid tumor to the exclusion criteria in Material and Methods.

Comment: 5) Page 8, regarding the correlation in figure 6, Authors are not allowed to draw such conclusion because an r of 0.4 is very low.

Answer: “Very weak” positive correlation is usually understood for r ranging 0-0.2 – our case, 0.4, is “weak/moderate”. We have changed the sentence:

“We observed a positive correlation between changes in urine 5-HIAA concentration and changes in serum concentration of 5-HT in SIBO-D patients after rifaximin treatment (p < 0.01, Figure 6).”

into:

“We observed a weak/moderate positive correlation between changes in urine 5-HIAA concentration and changes in serum concentration of 5-HT in SIBO-D patients after rifaximin treatment (p < 0.01, Figure 6).”

Comment: 6) Alteration of tryptophan pathways has been described in IBS patients as well.  Considering that in several cases IBS is related with SIBO, this observation should be analyzed and discussed.

Answer: We have added the following fragment to the Discussion section:

“Alterations of Trp metabolism pathways have been also described in IBS patients (reviewed in [22]). These alterations may be underlined by changes in the intestinal microflora, which metabolizes excessive dietary Trp and produces many Trp metabolites that are more active than the parental compound (reviewed in [23]). They may be linked with excessive immune activation of the host and progression/outcome of various diseases, including IBS (reviewed in [24]). SIBO is highly prevalent in IBS – its prevalence calculated in a recent meta-analysis was about 31% [25,26].  This meta-analysis included studies with different SIBO definitions and different methods to its diagnosis as well as different control groups – healthy subject and patients with non-IBD diseases. Our study shows an association between changes in Trp metabolism and SIBO occurrence – if such changes are associated with IBS, they could be still underlined by SIBO, unless the latter is excluded. To determine what is a primary reason of changes in Trp metabolism that are associated with several clinical symptoms requires additional cause-effects molecular studies.”

with the following new references:

  1. Agus, A.; Planchais, J.; Sokol, H. Gut Microbiota Regulation of Tryptophan Metabolism in Health and Disease. Cell host & microbe 2018, 23, 716-724, doi:10.1016/j.chom.2018.05.003.
  2. Berstad, A.; Raa, J.; Valeur, J. Tryptophan: 'essential' for the pathogenesis of irritable bowel syndrome? Scandinavian journal of gastroenterology 2014, 49, 1493-1498, doi:10.3109/00365521.2014.936034.
  3. Zhang, X.; Gan, M.; Li, J.; Li, H.; Su, M.; Tan, D.; Wang, S.; Jia, M.; Zhang, L.; Chen, G. Endogenous Indole Pyruvate Pathway for Tryptophan Metabolism Mediated by IL4I1. Journal of agricultural and food chemistry 2020, 68, 10678-10684, doi:10.1021/acs.jafc.0c03735.
  4. Shah, A.; Talley, N.J.; Jones, M.; Kendall, B.J.; Koloski, N.; Walker, M.M.; Morrison, M.; Holtmann, G.J. Small Intestinal Bacterial Overgrowth in Irritable Bowel Syndrome: A Systematic Review and Meta-Analysis of Case-Control Studies. The American journal of gastroenterology 2020, 115, 190-201, doi:10.14309/ajg.0000000000000504.
  5. Takakura, W.; Pimentel, M. Small Intestinal Bacterial Overgrowth and Irritable Bowel Syndrome - An Update. Frontiers in psychiatry 2020, 11, 664, doi:10.3389/fpsyt.2020.00664.

Reviewer 2 Report

I read with interest the MS "Serotonin pathway of tryptophan metabolism in the pathogen-esis and treatment of small intestinal bacterial overgrowth" by Chojnacki C and coworkers. The MS deals with an intriguing speculation, but a number of issues should be addressed, namely: a) the sensitivity and specificity of lactulose breath test to diagnose SIBO is controversial at best (J Neurogastroenterol Motil. 2020 Jan; 26(1): 16–28) as well as the role of Rifaximine on its management (Aliment Pharmacol Ther. 2017 Mar; 45(5): 604–616) for RCTs has not addressed the issue. My suggestion would be to tempere the Authors' enthusiam for their relevant data and to modify the whole MS starting from the title (eg Serotonin pathway correlates with lactulose breath hydrogen testing for a formal diagnosis of SIBO was not entertained), b) a formal diagnosis of either IBS-D or IBS-C according to Rome Criteria would be mostly welcomed if possible, c) there is no report of either associated bowel symptoms or Rifaximine side effects which are commonly reported by constipated patient when treated with any antibiotic, d) I suspect that an impaired small bowel transit plays a major role in the excessive breath H2 production in constipated patients, this should be dealed with both in the limitation and sicussion section, e) statements as the following "In conclusion, tryptophan metabolism may contribute to hydrogen-positive SIBO pathogenesis and it may influence the diversification of SIBO into variants with diarrhea and constipation. Urinary 5-HIAA concentration correlates with LHBT value, TPH-1 ex-pression in colonic mucosa and TH-5 in serum of SIBO patients, so it can be considered as at least dependent non-invasive marker of this disease" should be avoided for are not supported by the data which actually support additional randomized controlled trials.    

Author Response

I read with interest the MS "Serotonin pathway of tryptophan metabolism in the pathogen-esis and treatment of small intestinal bacterial overgrowth" by Chojnacki C and coworkers. The MS deals with an intriguing speculation, but a number of issues should be addressed, namely:

Comment: a) the sensitivity and specificity of lactulose breath test to diagnose SIBO is controversial at best (J Neurogastroenterol Motil. 2020 Jan; 26(1): 16–28)

Answer: We have added the following fragment to Discussion:

“In this study, we employed lactulose breath test, which recently was reported to have lower susceptibility, specificity and worse general performance as compared with its glucose counterpart on the basis of a meta-analysis [19]. However, lactulose test seems to be more reliable when intestinal motility is impaired [17,20]. Furthermore, glucose in contrary to lactulose, tends to be absorbed by the duodenum and jejunum, impeding microbiota assessment in distal parts of the small intestine.”

with new references:

  1. Losurdo, G.; Leandro, G.; Ierardi, E.; Perri, F.; Barone, M.; Principi, M.; Leo, A.D. Breath Tests for the Non-invasive Diagnosis of Small Intestinal Bacterial Overgrowth: A Systematic Review With Meta-analysis. Journal of neurogastroenterology and motility 2020, 26, 16-28, doi:10.5056/jnm19113.
  2. Pimentel, M. Breath Testing for Small Intestinal Bacterial Overgrowth: Should We Bother? The American journal of gastroenterology 2016, 111, 307-308, doi:10.1038/ajg.2016.30.

Comment:  as well as the role of Rifaximine on its management (Aliment Pharmacol Ther. 2017 Mar; 45(5): 604–616) for RCTs has not addressed the issue.

Answer: We have added the following sentence to Discussion:

“Although rifaximin was reported to be safe and effective in SIBO treatment in many studies, their quality is generally questioned, and well-planned randomized clinical trials are desired to verified these studies and establish a reliable treatment regime. “

Comment: My suggestion would be to tempere the Authors' enthusiam for their relevant data and to modify the whole MS starting from the title (eg Serotonin pathway correlates with lactulose breath hydrogen testing for a formal diagnosis of SIBO was not entertained),

Answer: Surely, as a formal diagnosis of SIBO has not been performed, we are not allowed to use the term SIBO, but it is our clinical practice. And we are not the only ones. Furthermore, hundreds of papers claim to deal with SIBO on the basis of breath, hydrogen or methane, tests only. It would be difficult for us to accept that the only thing we did was calculation of a correlation between results of a particular test with tryptophan metabolism without any clinical context.

We have modified the title to:

“Serotonin pathway of tryptophan metabolism in the pathogenesis and treatment of hydrogen-positive small intestinal bacterial overgrowth”

Comment: b) a formal diagnosis of either IBS-D or IBS-C according to Rome Criteria would be mostly welcomed if possible,

Comment: SIBO may be involved in several gastrointestinal disorder and we did not perform formal analyses of them, including IBS. We have added the following fragment to Discussion:

“Alterations of Trp metabolism pathways have been also described in IBS patients (reviewed in [22]). These alterations may be underlined by changes in the intestinal microflora, which metabolizes excessive dietary Trp and produces many Trp metabolites that are more active than the parental compound (reviewed in [23]). They may be linked with excessive immune activation of the host and progression/outcome of various diseases, including IBS (reviewed in [24]). SIBO is highly prevalent in IBS – its prevalence calculated in a recent meta-analysis was about 31% [25,26].  This meta-analysis included studies with different SIBO definitions and different methods to its diagnosis as well as different control groups – healthy subject and patients with non-IBD diseases. Our study shows an association between changes in Trp metabolism and SIBO occurrence – if such changes are associated with IBS, they could be still underlined by SIBO, unless the latter is excluded. To determine what is a primary reason of changes in Trp metabolism that are associated with several clinical symptoms requires additional cause-effects molecular studies.”

with the following new references:

  1. Agus, A.; Planchais, J.; Sokol, H. Gut Microbiota Regulation of Tryptophan Metabolism in Health and Disease. Cell host & microbe 2018, 23, 716-724, doi:10.1016/j.chom.2018.05.003.
  2. Berstad, A.; Raa, J.; Valeur, J. Tryptophan: 'essential' for the pathogenesis of irritable bowel syndrome? Scandinavian journal of gastroenterology 2014, 49, 1493-1498, doi:10.3109/00365521.2014.936034.
  3. Zhang, X.; Gan, M.; Li, J.; Li, H.; Su, M.; Tan, D.; Wang, S.; Jia, M.; Zhang, L.; Chen, G. Endogenous Indole Pyruvate Pathway for Tryptophan Metabolism Mediated by IL4I1. Journal of agricultural and food chemistry 2020, 68, 10678-10684, doi:10.1021/acs.jafc.0c03735.
  4. Shah, A.; Talley, N.J.; Jones, M.; Kendall, B.J.; Koloski, N.; Walker, M.M.; Morrison, M.; Holtmann, G.J. Small Intestinal Bacterial Overgrowth in Irritable Bowel Syndrome: A Systematic Review and Meta-Analysis of Case-Control Studies. The American journal of gastroenterology 2020, 115, 190-201, doi:10.14309/ajg.0000000000000504.
  5. Takakura, W.; Pimentel, M. Small Intestinal Bacterial Overgrowth and Irritable Bowel Syndrome - An Update. Frontiers in psychiatry 2020, 11, 664, doi:10.3389/fpsyt.2020.00664.

Comment: c) there is no report of either associated bowel symptoms or Rifaximine side effects which are commonly reported by constipated patient when treated with any antibiotic,

Answer: I am not sure whether I follow this remark. As we wrote in the manuscript, rifaximin was well tolerated and no side effects were observed in either SIBO group.

Comment: d) I suspect that an impaired small bowel transit plays a major role in the excessive breath H2 production in constipated patients, this should be dealed with both in the limitation and sicussion section,

Answer: We have added the following sentence to Discussion:

“Constipation is not a typical sign of SIBO and positive results in hydrogen test can be underlined by a delayed intestinal transit in constipation patients. However, the only reliable method to check the presence/absence of SIBO would be invasive analysis of the number of bacteria or/and composition of bacteria in aspiration fluid from the small intestine. This is a limitation of our study.”

Comment: e) statements as the following "In conclusion, tryptophan metabolism may contribute to hydrogen-positive SIBO pathogenesis and it may influence the diversification of SIBO into variants with diarrhea and constipation. Urinary 5-HIAA concentration correlates with LHBT value, TPH-1 ex-pression in colonic mucosa and TH-5 in serum of SIBO patients, so it can be considered as at least dependent non-invasive marker of this disease" should be avoided for are not supported by the data which actually support additional randomized controlled trials.

Answer: We have changed the questioned fragment into:

“In conclusion, the serotonin pathway of tryptophan metabolism may be changed in patients with SIBO diagnosed on the basis of lactulose breath test, but this should be confirmed by randomized clinical trials. As we showed that urinary 5-HIAA positively correlated with other Trp metabolites, it could be considered as a non-invasive marker in hydrogen-positive SIBO but, again, randomized clinical trials are needed to verify this speculation.”

Round 2

Reviewer 1 Report

Answers are ok

Author Response

Thank you.

Reviewer 2 Report

I read with interest the revised version of this valuable MS dealing with potential non-invasive markers of dybiosis. My queries have been partially addressed. I still consider the study as a pilot one for the diagnosis of SIBO is not thouroughly supported (eg constipated patients are included). The statement that is common clinical practice to diagnose SIBO by lactulose breath test is highly variable according to the setting. Moreover, no improvement on symptoms was searhced for after Rifaximin treatment to support the clinical relevance of the potential dysbiosis diagnosis. My personal suggestion would be to specify both in the title and in the MS that the study is a pilot one and additional RCTs are awaited to confirm the speculation. No additional suggestions on this side.

Author Response

Comment: I read with interest the revised version of this valuable MS dealing with potential non-invasive markers of dybiosis. My queries have been partially addressed. I still consider the study as a pilot one for the diagnosis of SIBO is not thouroughly supported (eg constipated patients are included). The statement that is common clinical practice to diagnose SIBO by lactulose breath test is highly variable according to the setting. Moreover, no improvement on symptoms was searhced for after Rifaximin treatment to support the clinical relevance of the potential dysbiosis diagnosis. My personal suggestion would be to specify both in the title and in the MS that the study is a pilot one and additional RCTs are awaited to confirm the speculation. No additional suggestions on this side.

Answer: We have made the following changes in response to the above comments:

Title

We changed: “Serotonin pathway of tryptophan metabolism in the pathogen-esis and treatment of hydrogen-positive small intestinal bacterial overgrowth”

into:

“Serotonin pathway of tryptophan metabolism in small intestinal bacterial overgrowth – a pilot study with patients diagnosed with lactulose hydrogen breath test and treated with rifaximin”

Discussion

We have added the following sentence:

“Further studies are required to confirm whether constipation can be a SIBO-associated symptom.”

We have changed the sentence:

“Although there is no generally accepted consensus on SIBO therapy, rifaximin is preferentially used [31].”

into:

“Although there is no generally accepted consensus on SIBO therapy, rifaximin is frequently used [31].”

We have added the following sentence at the very end of the section:

“Finally, we presented the results of studies on SIBO patients diagnosed with lactulose hydrogen breath test and treated with rifaximin, but our pilot study should be confirmed in RCTs.”